# The Occurrence of *Mycobacterium avium* Subspecies *paratuberculosis* Positive Milk Antibody ELISA Results in Dairy Cattle under Varying Time Periods after Skin Testing for Bovine Tuberculosis

**DOI:** 10.3390/ani11051224

**Published:** 2021-04-23

**Authors:** Nicole Bridges, Steven van Winden

**Affiliations:** Farm Animal Health and Production Group, Pathobiology and Population Sciences, Royal Veterinary College, Hawkshead Lane, Hatfield AL9 7TA, UK; svwinden@rvc.ac.uk

**Keywords:** Johne’s disease, paratuberculosis, milk ELISA, bovine tuberculosis test, dairy cattle

## Abstract

**Simple Summary:**

Milk from dairy cows is screened for antibodies to the bacterium causing Johne’s disease, which causes diarrhoea. The bacterium causing bovine tuberculosis has similarities in its structure with the bacterium causing Johne’s disease. Bovine tuberculosis is diagnosed by injecting avian and bovine tuberculin in the skin of the neck and comparing the reactions three days later. Potentially, tuberculosis testing may interfere with Johne’s disease testing. This study aimed to clarify the relationship between the time interval between tuberculosis and Johne’s disease testing and the number of Johne’s disease-positive cows. Data were collected from 51 herds, totalling 46,738 cow observations. Analysis showed that Johne’s disease testing in each 14 day interval increased the probability of detecting Johne’s disease-positive cows compared to Johne’s disease testing over 85 days after tuberculosis testing. The probability was 2.5-fold higher in the first 14 day interval after tuberculosis testing, increasing each two-week period to 4.0-fold higher at 57–70 days before dropping again. A previous history of tuberculosis within the herd increased the probability by 1.2 fold. This was less important compared to the timing of Johne’s disease testing after tuberculosis testing.

**Abstract:**

Enzyme-linked immunosorbent assays (ELISA) are used to screen cows for *Mycobacterium avium* subspecies *paratuberculosis* (MAP) infections, informing Johne’s disease (JD) management practices in dairy herds. The causative agent of bovine tuberculosis (bTB), *Mycobacterium bovis*, and MAP share multiple antigens. Moreover, *Mycobacterium avium* subspecies *avium* is used in the single intradermal cervical comparative tests (SICCT) that are routinely used in early detection of cows infected with bTB. Although these are different types of immune responses, potentially the SICCT may interfere with the levels of MAP antibodies. This study aimed to clarify the relationship between the SICCT-MAP milk ELISA testing interval and apparent prevalence of JD risk statuses. Data from 51 herds were used, totalling 46,738 cow observations. The Poisson models showed that MAP milk ELISA testing at 14 day intervals post-SICCT statistically significantly increased the odds of detecting JD-positive cows compared to JD testing 85+ days post-SICCT. The odds ratio (OR) started at 2.5 in the first 14 day interval post-SICCT, increasing each two-week period to an OR of 4.0 at 57–70 days, to subsequently drop. Additionally, a herd history of bTB increased the odds of detecting JD-positive cows (OR = 1.2); this was relatively limited compared to the magnitude of the post-SICCT effect.

## 1. Introduction

Johne’s disease (JD) is clinically defined by chronic diarrhoea due to a granulomatous enteritis in ruminants and the pathogen responsible for JD is *Mycobacterium avium* subspecies *paratuberculosis* (MAP) [1,2]. In dairy cattle, the herd economic consequences include shortened lifespans, restocking costs, lower milk yields, decreased feed conversion rate, fertility issues, reduced monetary returns at slaughter and higher prevalence of subclinical disease [3]. JD is also a significant welfare issue at the terminal stage, but in practice, most MAP-infected animals are sent for slaughter before reaching this point [1]. In order to reduce the impact on the dairy sector, JD control programmes are being put in place, for instance in Denmark and Canada [4,5]. In the United Kingdom (UK), the National Action Group on Johne’s aims to control JD in the dairy sector, coordinated by DairyUK and recently incorporated in the British dairy farm assurance scheme [6].

In these monitoring programmes, enzyme-linked immunosorbent assays (ELISA) in milk are used to screen cows at risk of being MAP infected as a cheap and quick alternative to faecal culture [7]. The sensitivity of these tests is limited, but with repeat tests every three months, this allows an assessment of the probability of MAP infection [8]. The repeat antibody testing allows categorising the cows as at ‘low’, ‘medium’ and ‘high’ risk of being affected by MAP infection and therefore the likelihood of shedding MAP in the faeces [9].

In the UK, another mycobacterium infection is present: *Mycobacterium bovis*, the causative agent of bovine tuberculosis (bTB) [10], is also a zoonosis and cattle in England are regularly tested using the single intradermal cervical comparative test (SICCT) [11,12,13]. The SICCT involves inoculation of *Mycobacterium bovis* and *Mycobacterium avium* subspecies *avium* purified protein derivatives (PPD) intradermally at two locations in the cervical region which are measured and compared at 72 h post-inoculation [14,15]. Cows reacting positive on SICCT are culled, herd movement restrictions are applied and a 60 day SICCT testing interval is introduced until the herd test repeatedly negative for bTB [16]. To regain Officially TB Free status, the whole herd must test negative on SICCT on two occasions within a short interval [17].

MAP and *Mycobacterium bovis* share multiple antigens [18]. *Mycobacterium bovis* antibody cross interference can reduce the specificity of MAP ELISA testing [19]. It is unlikely that intradermal injections with *Mycobacterium bovis* and *Mycobacterium avium* subspecies *avium* PPD result in a humoral (antibody) response. However, this cannot be fully dismissed either in sensitised cows as Casal et al. described an increased level of antibodies to *Mycobacterium bovis* 15 days after intradermal PPD injection and the *Mycobacterium avium* subspecies *avium* PPD component of the SICCT alone is likely to be sufficient to increase MAP milk ELISA titres due to an anamnestic response [20]. For this reason, National Milk Records (NMR) recommends to MAP milk ELISA test 42 days or more post-SICCT [21]. In an effort to measure the impact, May et al. showed the MAP milk ELISA titres at specific intervals over four months of 129 SICCT tested and 111 control animals [22]. They found that the MAP milk ELISA titres were significantly higher in the SICCT tested group compared to the control group when testing at 33 days post-SICCT. No significant difference was found at 53 days, 82 days or 110 days post-SICCT. They therefore recommended avoiding MAP milk ELISA testing until 53 days post-SICCT.

Similarly, Kennedy et al. compared the prevalence of MAP seropositive cows using milk and serum ELISA pre-SICCT and approximately every 14 days post-SICCT in a 139 cow herd [23]. Seroprevalence was 8% pre-SICCT, whilst 6% of the herd were positive on MAP milk ELISA pre-SICCT. They found MAP milk ELISA titres to be significantly higher at 30 days post-SICCT, and MAP serum ELISA titres at 58 days post-SICCT. No significant difference in milk ELISA titres was found at 43 days post-SICCT, or serum ELISA titres at 71 days post-SICCT. They therefore recommended postponing MAP milk ELISA testing for 43 days and MAP serum ELISA testing for 71 days post-SICCT. Similarly, Barden et al. showed an increased rate of a positive JD test result up to 30 days (odds ratio (OR) = 2.08) and 31–60 days (OR = 1.15) after bTB testing [24].

The current literature, however, is lacking more detailed time resolution as well as clarification on the importance of herds being infected with bTB in relation to the test (apparent) prevalence of JD-positive cows. In addition, it would be useful to know whether a farmer who has submitted an early JD test once would improve on their practices in subsequent submissions by ensuring a longer bTB–JD testing interval. This study aims to further clarify the relationship between the time interval between SICCT and MAP milk ELISA testing and the test prevalence of JD-positive cows. The objectives of this study were (1) to identify the odds for testing positive at 14 day bTB–JD testing interval increments, (2) to estimate the importance of being co-infected with bTB in relation to the odds for testing positive for JD on MAP milk ELISA and (3) to evaluate whether submitting once at a short bTB–JD testing interval has an effect on subsequent submission intervals. The authors anticipate that the findings will help farmers and veterinary surgeons to optimise the timing of their JD testing protocols.

## 2. Materials and Methods

Data were collected from a pool stored at National Milk Records, Wolverhampton. The JD test results were from herds testing between 1st January 2016 and 2nd February 2018. The milk pool consisted mostly of herds JD testing every quarter. Herds from the milk pool were selected to participate in this study based on two criteria to ensure data accuracy: (1) the herd must have exact SICCT and MAP milk ELISA testing dates and (2) the time interval between SICCT and MAP milk ELISA testing must be between 1 and 200 days inclusive. 

Each milk recording reported whether a cow was regarded as at ‘low’, ‘medium’ and ‘high’ risk of being affected by MAP infection. Testing for antibodies was performed by milk ELISA (IDEXX Paratuberculosis Screening Ab Test, IDEXX Laboratories, Westbrook ME, USA). The test interpretation followed the instructions of the manufacturer. Tests with a sample-to-positive ratio (S/P) of 0.3 or above were considered positive. Based on the repeat test results, cows were considered ‘low’ risk when tests returned negative or one positive out of the last three tests, but not the most recent test. ‘Medium’ risk cows returned alternating positive and negative results, or have the most recent test report positive and ‘high’ risk cows have repeatedly positive test results [25]. The SICCT consisted of PPDs of *Mycobacterium bovis* and *Mycobacterium avium* subspecies *avium.*

The following data were transferred to Microsoft Excel 2017: herd ID, herd size, date of SICCT, date of MAP milk ELISA, herd bTB status and numbers of ‘low’, ‘medium’ and ‘high’ risk cows. The bTB–JD testing interval was calculated as the number of days between SICCT and MAP milk ELISA testing. Based on the date of JD testing, year and quarter (Q) of testing was defined: Q1 consisted of tests from January to March, Q2 from April to June, Q3 from July to September, and Q4 from October to December. The bTB history was based on whether the farm was under bTB restrictions at any point in 2016.

### Statistical Analyses

Herd ID was considered a nominal categorical variable, as was quarter in the year. The bTB–JD testing interval was used as a continuous variable as well as grouped in an ordinal distribution of 14 day intervals: 0–14, 15–28, 29–42, 43–56, 57–70, 71–84, and 85 plus days. A short bTB–JD testing interval was considered to be shorter than 60 days. Year and herd size were deemed continuous variables, and bTB status (positive or negative) was considered a dichotomous categorical variable.

A mixed linear model was used, with herd ID as a random factor, to evaluate whether the bTB–JD testing interval varied per year, quarter, herd size, JD prevalence, or bTB status. A multivariate backward stepwise approach was performed to finalise an optimally descriptive model. Q–Q plots and normality of residuals were used to evaluate the models. In order to evaluate whether the bTB–JD testing interval was affected by an occurrence of a short (<60 day) testing interval, coded events of pre- and post-intervals were introduced to the final mixed linear model [24]. Finally, in case repeat events of a short bTB–JD testing interval occurred, these were coded and put in the final mixed linear model to enable evaluation of whether a repeat event had a different bTB–JD testing interval.

A mixed linear model with the Poisson distribution was used to evaluate the occurrence of a ‘medium’ or ‘high’ risk JD recording. Herd ID was a random effect and explanatory variables used were: year, quarter, herd size, ordinal categorised (14 day increments) bTB–JD testing intervals and bTB herd status. As the residuals were not normally distributed, one recording per herd was selected and the generalised linear model with a Poisson distribution was used with each farm represented in the data once. Which farm data point was used was based on a selection stratified on bTB–JD testing interval, with comparable observations in each category. Q–Q plots and normality of residuals were used to evaluate the models.

All the analyses were performed using IBM SPSS Statistics (Version 29, 2019 IBM Corp; Armonk, NY, USA) and an alpha error of less than 0.05 was considered statistically significant.

## 3. Results

Based on the selection criteria, data on 51 herds were obtained. The median herd size was 233, with an interquartile range (IQR) of 177 to 345. This totalled 46,738 cow observations, including multiple observations recorded from individual cows. Of the 46,738 cow observations, 41,769 were at ‘low’, 2563 at ‘medium’ and 2406 at ‘high’ risk of being affected by MAP infection. Per herd, there were on average 221 (median, IQR: 164–336), 12 (IQR: 5–22) and 9 (IQR: 5–21) in each respective risk category. 

Most herds (45/51; 88%) had repeat observations. The median number of recordings was 2 (IQR: 1–3). This provided a total of 163 herd observations. The mean bTB–JD testing interval was 100 days (95% confidence interval: 92–107) and the multivariate mixed linear model revealed that year and quarter were significantly associated with the length of bTB–JD testing interval. With progressing years, the bTB–JD testing interval on average increased by 24 days per year (*p* = 0.001); compared to the first quarter (January–March) the second quarter was 39 days longer (*p* < 0.001), the third quarter was 34 days longer (*p* = 0.002) and the fourth quarter was 14 days longer, which was not significant (*p* = 0.237).

In 31 of the herds, there was at least one recording with a bTB–JD test interval shorter than 60 days. In total, there were 38 short intervals, three herds each had a single repeat short (<60 day) bTB–JD testing interval, and two herds had two repeats each. The repeat short intervals were on average 12 days shorter than the initial short interval (average 39 days, *p* = 0.190). The bTB–JD testing interval was on average 6 days longer after a short interval, but this was not significant (*p* = 0.457). 

For each milk recording that reported a cow regarded as at ‘low’, ‘medium’ and ‘high’ risk of being affected by MAP infection, the individual stratified test resulted in a median (IQR) number of 212 (156–296), 9 (3–17) and 8 (4–5), respectively. In the time categories 0–14, 15–28 and 29–42, the number of reporting herds were: 5, 4 and 6, respectively. The other time periods all had 9 observations. The numbers of ‘medium’ and ‘high’ risk cows in each of the 14 day bTB–JD testing interval categories are depicted in Figure 1.

The final models for the ‘medium’ and ‘high’ risk cows had a respective R^2^ of 0.525 and 0.467. In both models, herd size, year, quarter, and bTB–JD testing interval were significant. In the ‘high’ risk cow model, a herd history of presence of bTB contributed to the number of cows testing positive for JD (*p* = 0.042). The results of the final multivariate Poisson model are shown in Table 1. 

Controlled for year, quarter and herd size, the odds of detecting ‘medium’ risk cows are significantly different in half of the testing interval categories, whereas the chance of detecting ‘high’ risk cows is significantly different in all of the testing interval categories. Additionally, the impact (OR estimates) is higher for the ‘high’ risk cows than the ‘medium’ risk cows, particularly given the lower overall prevalence of cows at ‘high’ risk of being affected by MAP infection.

For the original non-stratified dataset, the weighted average parameter estimate for the bTB–JD testing interval was 0.23 for ‘medium’ risk cows and 0.47 for ‘high’ risk cows, which equates to an additional 1.26 and 1.60 cows, respectively. Anticipating an extended bTB–JD testing interval of on average 24 additional days in the following year, this would equate to an additional 1.11 ‘medium’ risk and 1.26 ‘high’ risk cows, respectively.

## 4. Discussion

In the sample of herds, there are temporal variations that are similar with respect to the bTB–JD testing interval as well as the detection of JD-positive cows, labelled as at ‘medium’ and ‘high’ risk of being affected by MAP infection. This suggests that JD control is effective at driving the prevalence down, in line with the findings of Camanes et al. [26]. Part of the lowered apparent prevalence is explained by an extended bTB–JD testing interval: with on average an extra 24 days per year, the JD tests are less likely to be affected by the previous bTB test. Over the years, the odds for a positive JD test are declining, with the lowest apparent prevalence in the quarters that the cows are able to graze outside (quarter 2 and 3). The bTB–JD testing interval follows a similar trajectory; it is increasing over time and longer in summer months. As farmers are made aware of the importance of a bTB–JD testing interval of at least 42 days and preferably 60 days, they seem to get better at achieving this over time. There is, however, little evidence of learning from past events: when a farmer submits after a short bTB–JD testing interval, subsequent bTB–JD testing intervals are not necessarily extended. They were not likely to repeat the short bTB–JD testing interval, but the ones who do have a short interval of similar length suggest that the negative impact of a shortened interval is not fully recognised.

The Poisson regression results, however, do suggest that there is a diagnostic benefit in not JD testing too closely to the previous bTB test. This is in line with the observations of May et al. and Barden et al. [22,24]. The findings of the current study further clarify the effect of bTB–JD testing intervals: for ‘medium’ risk cows, the largest increase in probability of a positive JD test is with testing interval categories of 29–42 days (OR = 2.4) and 43–56 days (OR = 2.6). This is very similar to the findings of May et al., whilst Barden et al. reported an earlier (0–30 days post-bTB test) rise in odds for testing positive for JD [22,24]. The ‘high’ risk cows are more likely to be identified in the bTB–JD testing interval categories 43–56 and 57–70 days (OR = 3.2 and 4.0, respectively). The current study suggests that the relation of the previous bTB test to the JD test performance lasts for longer than previously thought. This is of particular importance for herds that have a history of bTB: the current study data show that this increases the chance of finding ‘high’ risk cows but also the farms are more likely to encounter bTB testing in close time proximity to JD testing, as they are testing at a shorter bTB testing interval (typically 60 days when an active infection is on the farm).

The finding of a higher positive test rate for JD in herds that have bTB is similar to what is reported by Picasso-Risso et al., with a similar OR of 1.47 [27]. A herd history of bTB raises the odds for finding ‘high’ risk cows (OR = 1.2) but also puts a ceiling on the maximum possible length of bTB–JD testing interval. The latter has a larger impact on detecting JD-positive cows based on the current study data (OR = 1.6). At a 60 day testing interval for bTB, this would put the bTB–JD testing interval in the 57–70 days bracket (OR = 4.0). To best avoid falsely detecting ‘high’ risk cows, there could perhaps be an argument for JD testing in the two weeks after bTB testing (OR = 2.5 for 0–14 days) rather than just before bTB testing, but further studies are required to investigate this. The authors speculate that this finding may be due to the time taken to mount a humoral (antibody) response. One would expect a continual rise in antibody production between day 1 and day 14, particularly in those cows infected with MAP who are therefore already sensitised to the antigen, with primed (memory) B cells enabling a more rapid humoral response. Further studies to investigate this prediction should analyse the dataset using the bTB–JD testing interval as a continuous variable, as opposed to grouped in an ordinal distribution of 14 day intervals. Further investigation of the relationship between the herd bTB results of the most recent bTB test and the numbers of ‘high’ risk cows would also be beneficial. In this study, the bTB history was based on whether the farm was under bTB restrictions at any point in 2016, so a more complete dataset providing herd bTB results for every bTB test would enable further evaluation. With this more detailed approach, the authors would expect a smaller *p*-Value and a greater effect size to be observed.

The question is whether this observed variation is due to an increased sensitivity of the MAP milk ELISA test, or whether more false positives are being detected due to a reduction in specificity with other cross-reacting antigens from related mycobacterial species. Natural infection with bTB elicits a Th1-type cell-mediated delayed hypersensitivity response [28]. The effect of PPD inoculation on bTB humoral immunity, responsible for cross reactions in MAP milk ELISA tests, is disputed, varying with antigen and stage of infection [29]. Interestingly, Kennedy et al. found a significant increase in MAP antibodies post-PPD administration [30]. This is in line with the findings of Casal et al. who found a significant increase in bTB antibodies after intradermal PPD injection during bTB testing [20]. The current study findings suggest that an increase in bTB antibodies not only lasts longer, but the odds of testing positive for JD also increase further as time progresses. This is similar to the response observed in bTB and the serologic response following PPD inoculation in infected cattle—the antibody rise is noticeable after two weeks, peaks in the next two weeks, and wanes but remains high the following weeks [31].

Equally, there could be *Mycobacterium avium* subspecies *avium* infections occurring during the housing period; this would explain a higher rate of ‘medium’ and ‘high’ risk cows in quarter 1 and 4, compared to the grazing period. Kennedy et al. proposed that cows housed overwinter are more likely to be exposed to mycobacterial antigens in avian droppings [23]. Thom et al., found that previous exposure to the environmental mycobacterium *Mycobacterium avium* subspecies *avium* increased IFN-gamma serum levels compared to control calves with no previous exposure [32]. Continual environmental exposure to various mycobacterial antigens such as *Mycobacterium avium* subspecies *avium* may also elicit a continual low level of plasma cell-mediated mycobacterial antibody production. The grazing period also coincides with early lactation for spring calving dairy herds. Higher milk yields during peak lactation have been shown to reduce MAP milk ELISA antibody titres due to an antibody dilution effect [33], which would explain a lower rate of ‘medium’ and ‘high’ risk cows during quarter 2 and 3.

In addition to the two reasons for a false-positive JD test result due to stimulation of the immune system (mycobacterial coinfection or bTB testing) reducing the specificity of the MAP milk ELISA, it could be that the sensitivity of the test is raised. This is specifically the case for ‘high’ risk cows, as this JD risk status requires repeat positive test results. This is particularly relevant for the proximity to the bTB test as the probability of multiple repeated positive test results being false positives is very low [8]. The ‘medium’ risk cows will require a single positive test, which is more easily influenced by a temporary reduction in specificity due to a recent bTB test. To be graded as a ‘high’ risk cow, however, requires repeated positive milk MAP ELISA tests, which is poorly explained by a one-off short bTB–JD testing interval. The observation that the association with the bTB–JD testing interval is more pronounced and more sustained suggests a possible increase in sensitivity of the MAP milk ELISA. Subsequent exploration of this notion was beyond the scope of the current study.

## 5. Conclusions

This study shows the impact of external events on the likelihood of detecting cows at ‘medium’ and ‘high’ risk of being affected by MAP infection. These external events are showing a decreased JD-positive case detection as the years progress and increased apparent prevalence of JD-positive cows over the housing period. The bTB–JD testing intervals showed similar trends, getting longer over the years and during grazing. Both suggest that the JD control programme is having an effect. The multivariate Poisson model also showed a sustained increase in detection of JD-positive cows in milk recordings that have a shorter bTB–JD testing interval. This is particularly the case for cows that are categorised as ‘high’ risk, as these cows have repeatedly tested positive, and are therefore likely to represent an increased sensitivity of the MAP ELISA test. More detailed, longitudinal studies are required to provide more definitive insight into the interaction of both tests used to control mycobacterial infections in the UK cattle population.

## Figures and Tables

**Figure 1 animals-11-01224-f001:**
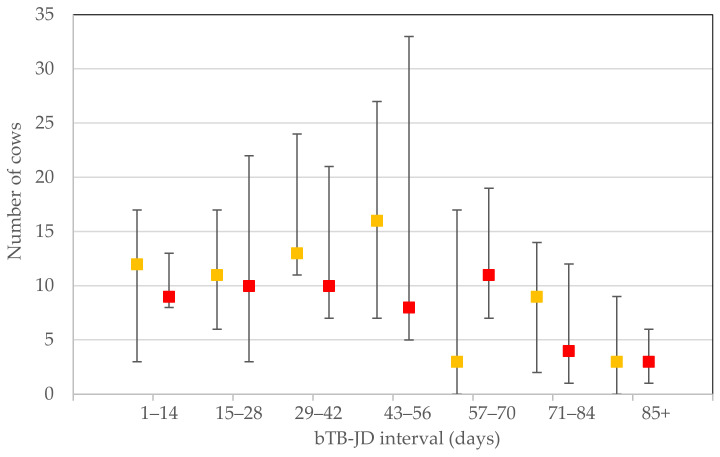
The number of ‘medium’ (amber) and ‘high’ (red) risk cows (median, interquartile range) reported for each bovine tuberculosis to Johne’s disease testing interval category.

**Table 1 animals-11-01224-t001:** Results of the multivariate Poisson regression model of cows testing positive for Johne’s disease using *Mycobacterium avium* subspecies *paratuberculosis* milk enzyme-linked immunosorbent assays.

Model Parameter	‘Medium’ Risk Cows		‘High’ Risk Cows	
	Odds Ratio (95% CI)	*p*-Value	Odds Ratio (95% CI)	*p*-Value
Herd size	1.002 (1.002–1.003)	<0.001	1.001 (1.001–1.002)	<0.001
Year	0.749 (0.599–0.938)	0.012	0.751 (0.594–0.949)	0.017
January–March	0.781 (0.562–1.087)	0.143	0.929 (0.664–1.300)	0.667
April–June	0.628 (0.506–0.779)	<0.001	0.364 (0.287–0.461)	<0.001
July–September	0.342 (0.249–0.468)	<0.001	0.528 (0.383–0.727)	<0.001
October–December	Reference		Reference	
Day 1–14	2.341 (1.580–3.468)	<0.001	2.453 (1.553–3.873)	<0.001
Day 15–28	1.197 (0.799–1.793)	0.384	2.510 (1.641–3.838)	<0.001
Day 29–42	2.401 (1.699–3.393)	<0.001	2.793 (1.853–4.209)	<0.001
Day 43–56	2.598 (1.878–3.594)	<0.001	3.329 (2.267–4.887)	<0.001
Day 57–70	1.337 (0.932–1.920)	0.115	3.960 (2.668–5.878)	<0.001
Day 71–84	1.438 (0.996–2.077)	0.053	2.019 (1.332–3.060)	0.001
Day 85+	Reference		Reference	
bTB history	-	NS	1.206 (1.006–1.444)	0.042

## Data Availability

Third-Party Data: Restrictions apply to the availability of these data. Data were obtained from National Milk Records and are available from the authors with the permission of National Milk Records.

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
