# Peer review of "The Occurrence of *Mycobacterium avium* Subspecies *paratuberculosis* Positive Milk Antibody ELISA Results in Dairy Cattle under Varying Time Periods after Skin Testing for Bovine Tuberculosis"

_animals, 2021, doi:10.3390/ani11051224_

Round 1

Reviewer 1 Report

In the manuscript the authors describe and anlayze the impact of the skintest, in particular the CCT or for old-fashioned people the SICCT (there is nothing single about the CCT). The analyses are thorough and the results of these analyses are presented clearly.

  • However, part of the introduction and definitely the discussion and conclusions are affected by overlooking the fact that the intradermal test has been in common use since the late 1980's in bTB diagnostics to enhance serology: a phenomenon called the anemnestic rise.
  • in particular, the presentation by Paul Wood at the first international conference on M. bovis (Vet. Mic. 40; 1-2, 125-135, 1994) dealing with the use of MPB70 in serology of bTB, sparked interest and made the use of the anamnestic rise an important part of the first EU project on blood based tests coordinated by Michael Monaghan.
  • there are numerous publications on using the anamnestic rise in bTB serology to enhance the sensitivity of either PPDb or the speicific antigens.
  • in fact: both ENFER and Idexx used this phenomenon to "sell" their assays as a replacement for the intradermal assays, ignoring the fact (fraud or ignorance?) that they were using post skin test sera collected a few weeks after to validate their assays. a fact overlooked at first by many potential users, including DEFRA/APHA and DAFM. In fact: without a prior skin test their assays would not have been useful at all, so replacing the intradermal assay was definitely a step too far.
  • in elephants and rhino's, skintest not being an option, the use of the anemnestic rise has also been shown to be useful
  • False positives do occur following accidental subcutaneous injection, this would be similar to vaccination..so not surprising but poorly documented.
  • for the paratb ELISA, the CCT would not be needed, so the authors should make the observation that only the PPDa component of the CCT should be sufficient to increase the titers in paratb serology.
  • for bTB serology: PPDb is essential since this contains bits and pieces of the specific antigens in a large variety of complexes: indicating that the anamnestic response in itself only increases the titers for specific antigens or in the case of paratb serology the titers for relevant antigens; MAP lacks dominant and specific excreted CMI antigens which are present in the supernatant of M. bovis cultures.
  • In the bTB literature there are also examples of the antibody response fading after the peak at 2-5 weeks post skin test and increasing again after the next skin test, similar to the observations made by the authors.
  • Obviously, the optimistc introduction in the UK of low risk areas with 2-3 yearly testing (never understood why this was accpetable to the EU given the high frequency of animal movement in the UK in combination with a very poor pre-movenent testing and post-movement follow up) is hampering the (annual) use of the anamnestic rise for enhancement of the serology of paratb. This added detail will improve explanation as well understanding of the observed difference between herds.
  • And allow a better identifcation of the moment of sampling as well as to be apart opf the education program of farmers and/or stakeholders.
  • This manuscript would benefit from taking the above into account, and by critical (re-) reading some of the literature on bTB serology, allowing a more adequate citation/background and discussion.
  • and probably most important: allow an optimal use of the analyses to be made in real life.

Reviewer 2 Report

The area of research that the authors have described has been an on-going but important issue with respect to Johne’s disease (JD) testing.  The presence of either co-infection with a related mycobacterium such as bovine tuberculosis (bTB) or Mycobacterium avium avium (Maa) or, PPDa/PPDb for  testing bovine TB has been demonstrated previously to affect the antibody tests for JD.  This is a timely and well constructed piece of research focussing in on the precise time duration after a bovine TB test (SICCT) that a JD antibody test result can be reliably interpreted.

However, while this manuscript has been well designed and written, there are some minor issues which the authors should consider undertaking, to improve the manuscript.

Please refer to the attached file for these suggested improvements.

Round 2

Reviewer 1 Report

Quite satisfied with the response and adjustments made by the authors!

Reviewer 2 Report

Thank you to the author's in their efforts to improve the manuscript.  In my opinion, all the comments and questions in my review have been addressed.